# Tailored Surgical Stabilization of Rib Fractures Matters More Than the Number of Fractured Ribs

**DOI:** 10.3390/jpm12111844

**Published:** 2022-11-04

**Authors:** Wen-Ruei Tang, Chao-Chun Chang, Chih-Jung Wang, Tsung-Han Yang, Kuo-Shu Hung, Chun-Hsien Wu, Yi-Ting Yen, Yau-Lin Tseng, Yan-Shen Shan

**Affiliations:** 1Department of Surgery, National Cheng Kung University Hospital, Tainan 704, Taiwan; 2Division of Trauma and Acute Care Surgery, Department of Surgery, National Cheng Kung University Hospital, Tainan 704, Taiwan

**Keywords:** thoracic trauma, surgical stabilization of rib fractures (SSRF), video-assisted thoracic surgery (VATS), tailored algorithmic approach, pulmonary complications

## Abstract

Background: Patients sustaining multiple rib fractures have a significant risk of developing morbidity and mortality. More evidence is emerging that the indication of surgical stabilization of rib fractures (SSRF) should expand beyond flail chest. Nevertheless, little is known about factors associated with poor outcomes after surgical fixation. We reviewed patients with rib fractures to further explore the role of SSRF; we matched two groups by propensity score (PS). Method: A comparison of patients with blunt thoracic trauma treated with SSRF between 2010 and 2020 was compared with those who received conservative treatment for rib fractures. Risk factors for poor outcomes were analyzed by multivariate regression analysis. Results: After tailored SSRF, the number of fractured ribs was not associated with longer ventilator days (*p* = 0.617), ICU stay (*p* = 0.478), hospital stay (*p* = 0.706), and increased nonprocedure-related pulmonary complications (NPRCs) (*p* = 0.226) despite having experienced much more severe trauma. In the multivariate regression models, lower GCS, delayed surgery, thoracotomy, and flail chest requiring mechanical ventilation were factors associated with prolonged ventilator days. Lower GCS, higher ISS, delayed surgery, and flail chest requiring mechanical ventilation were factors associated with longer ICU stays. Lower GCS and older age were factors associated with increased NPRCs. In the PS model, NPRCs risk was reduced by SSRF. Conclusions: The risk of NPRCs was reduced once ribs were surgically fixed through an algorithmic approach, and poor consciousness and aging were independent risk factors for NPRCs.

## 1. Introduction

Thoracic trauma is the second most common injury in major trauma [1,2]. Ribs are one of the most vulnerable structures and the number of fractured ribs has been shown to be proportional to the severity of thoracic trauma [2,3,4,5,6]. The chest wall is highly innervated by pairs of intercostal nerves. Patients with broken ribs secondary to traumatic insult suffer excruciating pain, which limits the ability to take deep breaths and expectorate and contributes to pulmonary complications such as atelectasis and pneumonia after thoracic trauma. Patients with multiple rib fractures, even without flail, have a significant risk of developing morbidity and mortality [2,7,8].

Introduction of multimodal analgesia and surgical stabilization has changed the landscape of rib fracture management. Despite limited evidence, multimodal analgesia has been shown to decrease opioid use, ventilator days, pulmonary complication incidence, and the length of both hospital and ICU stay [3,4,9,10,11,12]. Advances in regional anesthesia such as myofascial plane blocks and paravertebral blocks further broaden the options in pain control [13,14,15]. Similarly, more evidence is emerging that the indication of surgical stabilization of rib fractures (SSRF) should expand beyond flail chest [12,16,17,18]. Compared to conservative treatment, SSRF has been observed to be of benefit for reducing the patient-reported pain score, decreasing ventilator dependence, and shortening the length of ICU stay [19,20,21]. Nevertheless, little is known about the factors associated with poor outcomes in surgically fixed patients [16]. In addition, there is not enough discussion on whether the number of fractured ribs still matters after SSRF. Using the trauma register database in a tertiary trauma center, we retrospectively reviewed outcomes of patients with rib fractures. Firstly, we analyzed the impact of rib fracture numbers in the conservative treatment group (SSRF (−) group) and the tailored SSRF group (SSRF (+) group) in order to identify potential factors contributing to poor surgical outcomes. To further explore the role of SSRF, we matched the two groups by propensity score.

## 2. Materials and Methods

### 2.1. Patient Enrollment and Data Collection

The data were queried from the trauma registry database at our hospital which was prospectively maintained by 2 full-term trauma registries. Patients with blunt thoracic trauma admitted to our institution between January of 2010 and December of 2020 were reviewed for eligibility. Those with rib fractures undergoing SSRF were enrolled in the SSRF (+) group. Since conservative treatment has advanced a lot in recent years, such as multimodal analgesia, only patients who received conservative treatment between January of 2019 and December of 2020 were enrolled in the SSRF (−) group. A number of exclusion criteria included trauma victims without fractured ribs and patients who died within 48 h of severe trauma. Medical charts and surgical details were reviewed. Preoperative computed tomography (CT) and serial chest radiograms were reviewed till latest follow-up (at least 1 month postoperatively). All rib fractures were documented as their location and type based on the standard Müller AO (Arbeitsgemeinschaft für Osteosynthesefragen) classification system and the shaft part was further subdivided to anterolateral and posterolateral sections as proposed by Love et al. [22,23].

### 2.2. Algorithmic Approach with Tailored Surgical Planning

Algorithmic approach: As depicted in Figure 1, early SSRF is encouraged unless there is the presence of absolute contraindication such as unstable hemodynamics or severe traumatic brain injury with high potential to progress. For major trauma victims suffering multiple injuries, a sequence of surgeries should be tailored individually and determined by collaborative specialties. For patients with unstable spinal fractures, spine surgery is usually performed first to facilitate the following lateral decubitus position in thoracic surgery. For those with nonspine fractures, the order of surgery for different injuries is planned to minimize intraoperative position change and movement of the injured area, while facilitating the next surgical procedure without mutual interference. Temporary splinting is possible and concomitant fixation in one surgery is favored in our trust.

Surgical indication: in our institute, SSRF indications consist of flail chest, nonflail fractures with respiratory compromise or difficulty in weaning from ventilator, chest wall deformity or severely displaced fractures, thoracotomy for associated thoracic injury (e.g., hemothorax, lung laceration), and intractable pain despite appropriate analgesia.

Consideration for SSRF: Preoperative CT is obtained and carefully checked. Fixation site and rib plating number are considered case by case. In general, flail segments, severely displaced ribs, and ribs 3 to 8 are fixated in priority [17]. Ideally, both ends of flail segments are addressed [24]. All patients undergoing SSRF were generally anesthetized, and most of them were intubated with a single-lumen endotracheal tube. Double-lumen endotracheal intubation is preferred in patients with concomitant lung resection or eradication of lobulated hemothorax unless severe contralateral lung contusion or hypoxia is present. We usually place the patient in a lateral decubitus position with mild anterior tilt on a beanbag and the operative arm positioned at 100 degrees of flexion to maximally expose the surgical field. The pre-existing thoracostomy tube is removed to avoid hardware infection. The skin disinfection area covers sternum anterior and spinal process posterior. Video-assisted thoracoscopic (VATS) examination is introduced to facilitate fracture site localization or intrathoracic procedures. The incision and surgical approach are tailored according to the distribution of the fracture sites. A linear incision along the fracture sites is the option for patients with linearly arrayed single fracture on each rib. For fractures beneath the scapula, the incision would be made along the scapular border. The fixation could be accomplished using a right-angled drill and screwdriver with the scapula elevated by a Kocher retractor. For rib fractures close to the spine, the incision is made along the border of the erector spinae muscle, followed by dissection between the muscle and bony structure to expose the transverse process of spine. Another incision along the corresponding spinal process is made followed by the dissection between the muscle and the spinal process to facilitate fixation if necessary. For ribs with multiple and fragmented fractures, a posterolateral incision along the scapula is made for maximal exposure of the fracture sites.

A rehabilitation program is essential after a major trauma [25]. Physiotherapists are routinely consulted for all trauma patients in our Trauma team. However, it is up to the specialty of the physiotherapists to decide what kind of rehabilitation they need.

### 2.3. Outcomes

Outcomes included pulmonary complications and time-related parameters. Pulmonary complications consisted of nonprocedure-related pulmonary complications (NPRCs) and procedure-related pulmonary complications (PRCs). Adult respiratory distress syndrome (ARDS), pulmonary embolism, pneumonia, and pulmonary edema were considered as NPRCs. PRCs were defined as any event developing during the SSRF or resulting from implants, screws, and surgical wounds. Time-related outcomes comprised ventilator days, length of ICU and hospital stay, and time from trauma to ambulation.

### 2.4. Statistical Analyses

Numeric outcome variables were assessed by the Kolmogorov–Smirnov test for normality. Non-normally distributed numeric variables were presented as median (interquartile range, IQR) and were analyzed by the Mann–Whitney U test. Categorical variables were presented as counts (percentage). Pearson X2 and Fisher exact tests were applied to compare categorical variables. If the variable was skewed, natural log transformation was used before multiple linear regression analysis. In order to adjust for the complex interactions between variables, both multivariate regression and propensity score (PS) models were used.

Categorical outcomes were analyzed by multivariate logistic regression, followed by the Hosmer–Lemeshow test for goodness of fit. Those factors with *p*-value < 0.05 in univariate analysis were entered into multivariate analysis. Based on established factors associated with outcomes in the literature and key factors addressed in this study, fracture numbers and surgical approaches were also included in the analyses.

A PS matching was applied since the SSRF (+) group differed greatly from the SSRF (−) group. Among the variables considered in the PS model were sex, age, comorbidity, transfer from district hospitals, unilateral and bilateral rib fractures, number of rib fractures, concomitant intrathoracic, scapular, sternal, and clavicular injuries, flail chest requiring mechanical ventilation, trauma mechanism, injury severity score (ISS), and abbreviated injury scale (AIS). The Wilcoxon signed rank test was used. A *p* < 0.05 was considered to indicate significance. Statistical analyses were performed using SPSS Version 25.0 Windows (IBM SPSS Corp., Armonk, NY, USA).

## 3. Results

There were 1468 consecutive patients with rib fractures between 2010 and 2020. A total of 177 of them (12.1%) received rib plating with A Plus titanium implants (SSFR (+) group, 84 of them receiving SSRF between 2019 and 2020). A total of 1291 patients received conservative treatment, 256 of whom were admitted between 2019 and 2020 and were enrolled as SSFR (−) group (Figure 2). All patients were followed for at least 1 month after hospital discharge. Median (IQR) follow-up time was 18.1 (5.2–29.4) months. Appendix A provides unmatched demographics and preoperative variables between the SSRF (−) and the SSRF (+) groups. Outcomes of unmatched groups are provided in Appendix A.

Presented in Table 1, for those who receiving SSRF, 106 of the 177 patients (59.9%) were male, and the median (IQR) age was 56.0 (45.5–64.5) years. Around one-third of patients were transferred from district hospitals. Presented in Table 2, road traffic accident (blunt trauma) was the most common cause. The median (IQR) ISS was 17 (13–24). The median (IQR) number of ribs fractured was 7 (5–9) and two-thirds of the fractures involved the posterolateral section (PL). In our cohort, flail chest presented radiologically in 72 patients (40.7%), though there was only 26 of them requiring mechanical ventilation support. The median (IQR) time from trauma to surgical fixation was 4.0 (2.5–7.0) days with a median (IQR) of 4 (3–5) plated ribs. Around 70% of patients underwent SSRF using video-assisted minithoracotomy through single lung ventilation. Presented in Table 3, the median (IQR) postoperative ventilator days were 2.0 (1.0–4.0) days and 0.0 (0.0–1.0) days for those with respiratory compromised and for all cohorts, respectively. The median (IQR) ICU stay and postoperative ICU stay were 6.0 (3.0–12.0) days and 4.0 (2.0–7.0) days, respectively. Patients sat out of bed with a median (IQR) of 3.0 (1.0–7.5) days. For PRCs, there were 2 hardware failures (1.1%) and 2 surgical site infections (1.1%, one of them even progressed to chest wall abscess and empyema). Pneumonia occurred in 13 patients (7.3%). One patient developed ARDS preoperatively and one patient had pulmonary edema after the surgery. There was no mortality or need for prolonged ventilation support.

As demonstrated in Figure 3, the PL to anterolateral section (AL) of the third to eighth ribs were most often involved, whereas when the first and second ribs were injured, the PL to posterior section (*p*) was the most vulnerable. Additionally, the first and second rib injuries were frequently associated with clavicle (81.0% vs. 36.3%; *p* < 0.01), sternum (14.3% vs. 1.5%; *p* < 0.01), and scapula fractures (33.3% vs. 17.0%; *p* = 0.02). The fourth to eighth ribs were surgically fixated most often.

Subgroup analysis of the SSRF (+) group is provided in Appendix A. Age, gender, comorbid conditions, and trauma mechanism did not differ significantly between the more than six fractured ribs and no more than six fractured ribs groups. Shown in Table 4, patients with more than six fractured ribs had longer ventilator days, length of ICU stay, hospital stay, and time from trauma to ambulation (*p* < 0.01, respectively). However, the number of rib fractures was not an independent factor for ventilator days (*p* = 0.617), ICU stay (*p* = 0.478), hospital stay (*p* = 0.706), and NPRCs (OR, 0.734; 95% CI, 0.444–1.211; *p* = 0.226) after SSRF in multivariate models despite having experienced much more severe trauma (higher ISS (21.5 vs. 17.0; *p* < 0.01), more bilateral rib fractures (21.7% vs. 5.9%; *p* < 0.01), more flail segments (62.0% vs. 17.6%; *p* < 0.01), and more associated lung parenchyma injuries (47.8% vs. 28.2%; *p* < 0.01)). Details of univariate analyses of the SSRF (+) group are provided in Appendix A. In the multivariate regression models, longer ventilator days was associated with lower initial GCS (*p* = 0.028), initial flail chest requiring mechanical ventilation (*p* = 0.020), longer time from trauma to SSRF (*p* = 0.025), and thoracotomy approach (*p* = 0.038) (Table 5). Longer ICU stay was associated with poorer GCS (*p* = 0.008), higher ISS score (*p* = 0.003), flail chest requiring mechanical ventilation (*p* < 0.001), and longer time from trauma to SSRF (*p* = 0.007) (Table 6). Poorer GCS (*p* = 0.037), higher ISS score (*p* < 0.001), and flail chest requiring mechanical ventilation (*p* = 0.049) were independent factors associated with longer hospital stay (Table 7). Presented in Table 8, risk factors for NPRCs were older age (OR, 1.097; 95% CI, 1.025–1.173; *p* = 0.007) and lower GCS (OR, 0.648; 95% CI, 0.499–0.843; *p* = 0.001). The Hosmer–Lemeshow test suggested the logistic model was accurate (χ2 = 1.757; *p* = 0.99).

For those receiving conservative treatment, the number of fractured ribs was associated with longer ICU stay (*p* = 0.029) in the multivariate model, but not with ventilator days (*p* = 0.282), hospital stay (*p* = 0.861), and NPRCs (*p* = 0.418). Details of univariate analyses of the SSRF (−) group are provided in Appendix A. Multivariate model results of ventilator days, ICU stay, hospital stay, and NPRCs are provided in Appendix A, respectively.

Table 9 provides PS matched demographic and perioperative variables. Table 10 shows that matched SSRF (+) patients sat up earlier (*p* = 0.002) and had lower NPRCs (17.0% vs. 6.6%; *p* = 0.019) despite not having shorter ventilator days, ICU stays, and hospital stays. Although not statistically significant, SSRF reduced the need for tracheostomies as well (3.8% vs. 0.0%; *p* = 0.121).

## 4. Discussion

Conservative treatment of multiple rib fractures has been associated with high incidence of pulmonary complications, prolonged ICU and hospital length of stay, and persistence of chronic pain [2,26,27,28,29,30]. Without surgical intervention, the overall pulmonary complications and mortality ranged between 35% and 48%, and 3% and 18%, respectively [2,3,4,8]. Compared to conservative treatment, SSRF has been shown to improve short-term outcomes and quality of life with flail chest being most extensively studied [19,20,21,26,29]. Both our multivariate regression and PS models did not predict shorter ventilator days, shorter ICU stays, or shorter hospital stays. Despite this, patients who received SSRF sat out of bed earlier (*p* = 0.002) and had lower NPRCs (*p* = 0.019). A decrease in tracheostomy requirements was observed without significance, which might have been due to a small number of cases.

An increased number of fractured ribs has been directly associated with the severity of thoracic injury and higher pulmonary morbidity and mortality [3,4,5]. Flagel et al. demonstrated that mortality rate and incidence of pulmonary complications increased significantly when six or more ribs were fractured, so did length of hospital stay for up to seven rib fractures [4]. We found that the number of fractures was correlated with longer ICU stays (*p* = 0.029) in the SSRF (−) group. However, in the SSRF (+) group, the number of fractures was not associated with ventilator days (*p* = 0.617), length of ICU stay (*p* = 0.478), hospital stay (*p* = 0.706), and NPRCs (OR, 0.734; 95% CI, 0.444–1.211; *p* = 0.226) after adjusting for age, comorbid conditions, and injury severity. Furthermore, overall NPRCs (7.3%) after SSRF was low in our cohort as compared to both the SSRF (−) group (18.8%; *p* < 0.01) and previous studies without SSRF. It could also provide indirect evidence of the benefit of SSRF for decreasing NPRCs due to early ICU liberation regardless of the number of broken ribs. Nevertheless, it is noteworthy that, without being adjusted, length of hospital stay was still longer in the group with more than six rib fractures despite comparable ventilator days and ICU stay. We later adjusted the numbers of fractured ribs with both a multivariable regression model (Table 7) and a propensity score model (Table 10) to show that after surgery fixation, the number of fractured ribs is no longer important, as our title states. This suggested that the number of fractured ribs correlated with the frequency of concomitant lung parenchyma injury (47.8% vs. 28.2%; *p* < 0.01) and extrathoracic injuries (extremity AIS ≥ 3: 21.7% vs. 4.7%; *p* < 0.01) and was an indicator of overall trauma severity (ISS: 21.5 vs. 17.0; *p* < 0.01) resulting in prolonged hospital stay at first glance despite leaving the ICU early. However, after adjustment, the number of rib fractures was not an independent variable of these outcomes.

While the results of SSRF seem positive, the optimal approach is still under debate. Single lung ventilation is advocated by most surgeons because of better exposure and less risk of lung parenchyma injury [17]. Our cohort was in line with this idea with around 70% of patients receiving single lung ventilation. This choice was independent of the number of fractured ribs (*p* = 0.922) and did not affect operating time (*p* = 0.242) but was correlated with ISS (single vs. double lung ventilation, 17.0 vs. 19.5; *p* = 0.047). In the situation of hemoptysis, double lumen intubation is required to maintain a patent airway in advance and to prevent soiling the contralateral lung. Nevertheless, due to increased intrapulmonary right-to-left shunt, pre-existing lung diseases should be evaluated carefully. With the aid of CT images and proficiency in SSRF, preoperative rib thickness measurements and temporary lowering tidal volume or apnea while placing screws is another feasible choice whenever one encounters intolerance of one lung ventilation. Likewise, double lumen intubation is not without risk. Postoperative hoarseness and bronchial injuries had been observed [31]. Furthermore, double lumen intubation is much more technically demanding, time consuming, and requires replacement with a single lumen tube postoperatively once longer ventilation support is necessary. In case of emergency, single lumen intubation followed by thoracotomy is necessary. In our cohort, patients undergoing thoracotomy were almost twice as likely as those receiving video-assisted surgery to have single lumen intubation (40.5% vs. 22.8%; *p* = 0.026).

Thoracotomy causes destruction of the already injured chest wall. Even modified by muscle-sparing technique, the larger incision of thoracotomy has been shown to increase morbidity and mortality. Since the introduction of video-assisted thoracoscopic surgery (VATS), VATS has been gaining popularity in different aspects of thoracic surgery because of less invasiveness and better postoperative recovery [32]. In our cohort, thoracotomy was associated with longer ventilator days (*p* = 0.038) but not ICU stay (*p* = 0.265) after adjustment for comorbidity and ISS in multiple linear regression models. This difference emphasized the great influence of thoracotomy on the integrity of chest wall muscles for weaning from the ventilator. Therefore, video-assisted minithoracotomy was often used in our institute unless contraindicated or the presence of clear indication for thoracotomy. Vertical incision along the medial border of the latissimus dorsi over the epicenter of the fracture site and one camera port at 7th intercostal space were made. As depicted in Figure 3, the third to eighth ribs were involved most often [17,33]. We plated ribs fourth to eighth in priority because they contribute the majority of rib cage stability and generate the most pain [12]. Furthermore, rib fractures most often involved PL (65.9%), an anatomical weak point and the point of impact loading, followed by AL (23.0%), which was consistent with the findings of Liebsch et al. [33]. Therefore, posterior vertical incision sufficed most of the time. Dissecting beneath muscle fascial planes anteriorly and posteriorly, AL and *p* injuries could be fixated by means of thoracoscopic localization and right-angled devices.

Our present study faced several limitations such as relatively small sample size, and a single-center experience. Because SSRF is not covered by national health insurance in our country, each rib plate costs around 2000 USD, resulting in selection bias. Consequently, many patients might not receive SSRF due to price rather than severity, which also impacts the choice of appropriate control group. In addition, some victims were not covered by private health insurance, which greatly limited the appropriate number of stabilizations. Patient-reported outcomes were not measured and long-term benefits or complications of SSRF could not be evaluated on account of a lack of chronic pain assessment. Further studies with a larger sample size are needed to find prognostic factors as to better define those who benefit most from SSRF.

## 5. Conclusions

The risk of NPRCs was reduced through an algorithmic approach. Once surgically fixated through a personalized approach, the number of fractured ribs, even more than six ribs, was not associated with longer ventilator days, length of ICU stay, length of hospital stay, and NPRCs. Poor consciousness and older age were risk factors for NPRCs independent of the number of rib fractures.

## Figures and Tables

**Figure 1 jpm-12-01844-f001:**
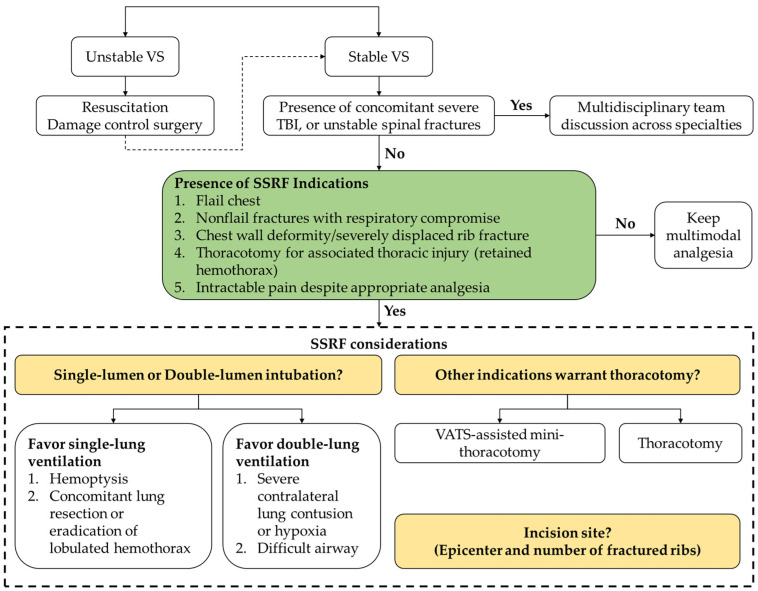
SSRF flowchart at our institute; TBI, traumatic brain injury; VAT, video-assisted thoracic surgery; VS, vital signs.

**Figure 2 jpm-12-01844-f002:**
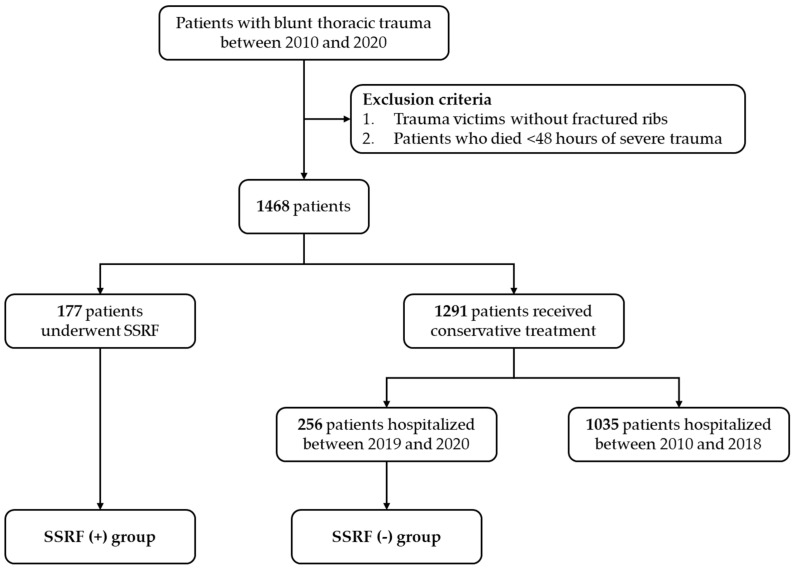
Flow chart of study enrollment. SSRF, surgical stabilization of rib fractures.

**Figure 3 jpm-12-01844-f003:**
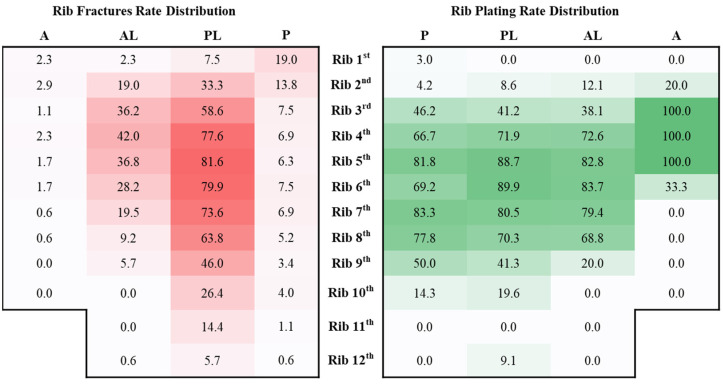
Rib fractures rate distribution (red heatmap): calculated as bilateral fracture sites per rib divided by person; rib plating rate distribution (green heatmap): calculated as bilateral plates number per rib divided by total fracture sites per rib; A, anterior section; AL, anterolateral section; PL, posterolateral section; P, posterior section.

**Table 1 jpm-12-01844-t001:** Demographics of SSRF (+) group.

	(N = 177)
Age at time of surgery, y	56.0 (45.5–64.5)
Sex (male)	106 (59.9)
Transferal	59 (33.9)
Charlson Comorbidity Index	2.0 (0.0–3.0)
Medication required hypertension	51 (28.8)
Coronary artery disease	13 (7.3)
Atrial fibrillation	5 (2.8)
Peripheral artery disease	2 (1.1)
Heart failure (ACC/AHA ≥ stage C)	1 (0.6)
Medication required diabetes mellitus	37 (20.9)
Chronic kidney disease (eGFR < 60 mL/min)	5 (2.8)
Liver cirrhosis	3 (1.7)
Connective tissue disease	3 (1.7)
Chronic obstructive pulmonary disease/Asthma	2 (1.1)
Psychiatric disorders	7 (4.0)
Malignancy	11 (6.2)

SSRF, surgical stabilization of rib fractures.

**Table 2 jpm-12-01844-t002:** Perioperative variables of SSRF (+) group.

	2010–2018 N = 93	2019–2020 N = 84	2010–2020 N = 177
Glasgow Coma Scale	15 (15–15)	15 (15–15)	15 (15–15)
Trauma mechanism			
Vehicle-to-vehicle collision	60 (64.5)	52 (61.9)	112 (63.3)
Vehicle-to-pedestrian collision	3 (3.2)	3 (3.6)	6 (3.4)
Single-vehicle collision	15 (16.1)	17 (20.2)	32 (18.1)
Fall	15 (16.1)	11 (13.1)	26 (14.7)
Crush	0 (0.0)	1 (1.2)	1 (0.6)
Injury Severity Score	20 (14–25)	17 (13–22)	17 (13–24)
Head/Neck			
Median (IQR)	0 (0–2)	0 (0–2)	0 (0–2)
AIS ≥ 3	16 (17.2)	14 (16.7)	30 (16.9)
Face			
Median (IQR)	0 (0–0)	0 (0–0)	0 (0–0)
AIS ≥ 3	1 (1.1)	0 (0.0)	1 (0.6)
Thorax			
Median (IQR)	3 (3–4)	3 (3–3)	3 (3–3)
AIS ≥ 3	93 (100.0)	84 (100.0)	177 (100.0)
Abdomen			
Median (IQR)	0 (0–2)	0 (0–2)	0 (0–2)
AIS ≥ 3	15 (16.1)	11 (13.1)	26 (14.7)
Extremity			
Median (IQR)	2 (0–2)	2 (0–2)	2 (0–2)
AIS ≥ 3	15 (16.1)	9 (10.7)	24 (13.6)
External			
Median (IQR)	0 (0–0)	0 (0–0)	0 (0–0)
AIS ≥ 3	1 (1.1)	1 (1.2)	2 (1.1)
Pattern of rib fractures			
Number of ribs broken	7 (5–9)	6 (5–8)	7 (5–9)
Presence of a flail segment radiologically	37 (39.8)	35 (41.7)	72 (40.7)
Requiring mechanical ventilation	15 (16.1)	11 (13.1)	26 (14.7)
Fractured side			
Right	34 (36.6)	35 (41.7)	69 (39.0)
Left	43 (46.2)	40 (47.6)	83 (46.9)
Both	16 (17.2)	9 (10.7)	25 (14.1)
Position on involved ribs (total fractured site = 1502)			
Anterior	2 (0.2)	21 (3.1)	23 (1.5)
Anterolateral	209 (25.5)	137 (20.1)	346 (23.0)
Posterolateral	534 (65.1)	456 (66.9)	990 (65.9)
Posterior	75 (9.1)	68 (10.0)	143 (9.5)
Associated injuries			
Lung contusion/laceration	40 (43.0)	28 (33.3)	68 (38.4)
Pneumothorax	51 (54.8)	43 (51.2)	94 (53.1)
Hemothorax	81 (87.1)	70 (83.3)	151 (85.3)
Cardiac injury	2 (2.2)	0 (0.0)	2 (1.1)
Great vessels	0 (0.0)	1 (1.2)	1 (0.6)
Soft tissue	21 (22.6)	22 (26.2)	43 (24.3)
Concurrent sternal fracture	4 (4.3)	4 (4.8)	8 (4.5)
Concurrent ipsilateral clavicular fracture	38 (40.9)	45 (53.6)	83 (46.9)
Concurrent ipsilateral scapular fracture	19 (20.4)	18 (21.4)	37 (20.9)
Surgery-related factors			
Indication			
Flail chest	37 (39.8)	35 (41.7)	72 (40.7)
nonflail fractures with respiratory compromise	8 (8.6)	6 (7.3)	14 (7.9)
Chest wall deformity/severely displaced	5 (5.4)	6 (7.3)	11 (6.2)
Thoracotomy for associated thoracic injury (e.g., hemothorax, lung laceration)	77 (82.8)	42 (50.0)	119 (67.2)
Intractable pain despite appropriate analgesia	58 (62.4)	61 (74.4)	119 (67.2)
Time from trauma to rib fixation, d	4.0 (2.0–6.0)	3.0 (1.3–5.0)	4.0 (2.5–7.0)
Number of surgically fixated rib fractures	4 (3–5)	4 (3–5)	4 (3–5)
Fracture fixation ratio	0.6 (0.4–0.8)	0.6 (0.5–0.8)	0.6 (0.5–0.8)
Mechanical ventilation during anesthesia (N = 169)			
Single lung ventilation	69 (80.2)	54 (65.1)	123 (69.5)
Double lung ventilation	17 (19.8)	29 (34.9)	46 (26.0)
Approach side			
Right	46 (49.5)	40 (47.6)	86 (48.6)
Left	45 (48.4)	44 (52.4)	89 (50.3)
Both	2 (2.2)	0 (0.0)	2 (1.1)
Incision			
Thoracotomy	27 (29.0)	21 (25.0)	48 (27.1)
Video-assisted minithoracotomy	66 (71.0)	63 (75.0)	129 (72.9)
Combined operation	45 (48.4)	45 (53.6)	90 (50.8)
Operation time, min	144 (123–204)	154 (125–198)	145 (124–198)
Blood loss, mL	100 (0–200)	100 (0–188)	100 (0–200)

Flail chest, the presence of three or more contiguous ribs fractured in two or more places; fracture fixation ratio, total fixated ribs divided by total fractured ribs; SSRF, surgical stabilization of rib fractures.

**Table 3 jpm-12-01844-t003:** Outcome variables of SSRF (+) group.

	N = 177
Ventilator days, d (N = 70)	5.0 (2.0–9.0)
Postoperative ventilator days, d (N = 70)	2.0 (1.0–4.0)
Length of ICU stay, d (N = 93)	6.0 (3.0–12.0)
Postoperative ICU stay, d (N = 93)	4.0 (2.0–7.0)
Length of hospital stay, d	13.0 (9.0–19.5)
Postoperative hospital stay, d	8.0 (5.0–13.0)
Time from trauma to ambulation, d	3.0 (1.0–7.5)
Time from rib fixation to ambulation, d	2.0 (1.0–4.0)
Procedure-related complication	
Surgical site infection/chest wall abscess	2 (1.1)
Empyema	1 (0.6)
Implant breakage	1 (0.6)
Implant dislodgement	1 (0.6)
Nonprocedure-related pulmonary complication	13 (7.3)
Tracheostomy	0 (0.0)
Mortality	0 (0.0)

**Table 4 jpm-12-01844-t004:** A comparison of the outcomes of SSRF (+) group with and without more than six fractured ribs.

	Fractured Ribs ≤ 6 N = 85	Fractured Ribs > 6 N = 92	*p*
Ventilator days, d	0.0 (0.0–0.0)	1.0 (0.0–6.0)	<0.01
Postoperative ventilator days, d	0.0 (0.0–0.0)	0.0 (0.0–2.0)	<0.01
Length of ICU stay, d	0.0 (0.0–3.0)	4.0 (0.0–11.0)	<0.01
Postoperative ICU stay, d	0.0 (0.0–2.0)	2.0 (0.0–5.0)	<0.01
Length of hospital stay, d	10.0 (7.7–15.5)	15.0 (11.0–21.8)	<0.01
Postoperative hospital stay, d	6.0 (4.5–10.0)	9.0 (7.0–16.0)	<0.01
Time from trauma to ambulation, d	2.0 (1.0–5.0)	4.0 (2.0–10.0)	<0.01
Time from rib fixation to ambulation, d	2.0 (1.0–3.0)	2.0 (1.0–5.8)	<0.01
Procedure-related complication			NA
Surgical site infection/chest wall abscess	0 (0.0)	2 (2.2)	
Empyema	0 (0.0)	1 (1.1)	
Implant breakage	1 (1.2)	0 (0.0)	
Implant dislodgement	1 (1.2)	0 (0.0)	
Nonprocedure-related pulmonary complication	3 (3.5)	10 (10.9)	0.06
Tracheostomy	0 (0.0)	0 (0.0)	NA
Mortality	0 (0.0)	0 (0.0)	NA

NA, not applicable

**Table 5 jpm-12-01844-t005:** Multiple linear regression analysis for natural log-transformed ventilator day.

Variables	Unstandardized Coefficients		Standardized Coefficients Beta	t	Sig.	95% CI for B	
	B	Std. Error				Lower	Upper
GCS	−0.072	0.032	−0.243	−2.264	0.028	−0.135	−0.008
ISS	0.012	0.009	0.164	1.410	0.164	−0.005	0.030
Flail chest requiring mechanical ventilation	0.492	0.206	0.271	2.387	0.020	0.079	0.904
Time from trauma to SSRF	0.053	0.023	0.248	2.302	0.025	0.007	0.099
Video-assisted vs. Thoracotomy	−0.438	0.206	−0.238	−2.122	0.038	−0.852	−0.024
Blood loss during SSRF	0.001	<0.001	0.196	1.700	0.095	<0.001	0.001
Number of ribs broken	0.018	0.035	0.059	0.502	0.617	−0.053	0.088

CI, confidence interval; GCS, Glasgow coma scale; ISS, injury severity score; SSRF, surgical stabilization of rib fracture. All variables with *p* < 0.05 in univariate analysis are included in a multivariate regression model of natural log-transformed ventilator day. Number of ribs broken is included because it is the target variable in the present study.

**Table 6 jpm-12-01844-t006:** Multiple linear regression analysis for natural log-transformed length of ICU stay of SSRF (+) group.

Variables	Unstandardized Coefficients		Standardized Coefficients Beta	t	Sig.	95% CI for B	
	B	Std. Error				Lower	Upper
GCS	−0.075	0.027	−0.225	−2.736	0.008	−0.129	−0.020
ISS	0.022	0.007	0.279	3.110	0.003	0.008	0.037
Bilateral vs. unilateral rib fractures	0.379	0.196	0.177	1.935	0.057	−0.011	0.768
Flail chest requiring mechanical ventilation	0.680	0.171	0.344	3.984	<0.001	0.340	1.019
Lung contusion/laceration	0.123	0.150	0.074	0.825	0.412	−0.174	0.421
Time from trauma to SSRF	0.052	0.019	0.223	2.784	0.007	0.015	0.089
Video-assisted vs. Thoracotomy	−0.177	0.158	−0.094	−1.121	0.265	−0.492	0.137
Blood loss during SSRF	<0.001	<0.001	0.082	0.948	0.346	<0.001	0.001
Number of ribs broken	−0.021	0.030	−0.070	−0.713	0.478	−0.080	0.038

CI, confidence interval; GCS, Glasgow coma scale; ISS, injury severity score; SSRF, surgical stabilization of rib fracture. All variables with *p* < 0.05 in univariate analysis are included in a multivariate regression model of natural log-transformed length of ICU stay.

**Table 7 jpm-12-01844-t007:** Multiple linear regression analysis for natural log-transformed length of hospital stay of SSRF (+) group.

Variables	Unstandardized Coefficients		Standardized Coefficients Beta	t	Sig.	95% CI for B	
	B	Std. Error				Lower	Upper
GCS	−0.049	0.023	−0.145	−2.107	0.037	−0.096	−0.003
ISS	0.023	0.005	0.366	4.636	<0.001	0.013	0.032
Bilateral vs. unilateral rib fractures	0.012	0.117	0.007	0.104	0.917	−0.219	0.243
Presence of a flail segment radiologically	0.095	0.089	0.080	1.071	0.286	−0.081	0.271
Flail chest requiring mechanical ventilation	0.247	0.125	0.144	1.982	0.049	0.001	0.493
Lung contusion/laceration	0.036	0.085	0.030	0.431	0.667	−0.131	0.204
Time from trauma to SSRF	0.019	0.010	0.117	1.873	0.063	−0.001	0.039
Video-assisted vs. Thoracotomy	−0.052	0.092	−0.039	−0.571	0.569	−0.234	0.129
Double lumen vs. Single lumen intubation	−0.083	0.083	−0.064	−1.005	0.316	−0.247	0.080
Number of surgically fixated rib fractures	0.022	0.033	0.057	0.670	0.504	−0.043	0.087
Operation time	0.001	0.001	0.112	1.370	0.173	<0.001	0.003
Blood loss during SSRF	<0.001	<0.001	0.037	0.521	0.603	<0.001	<0.001
Number of ribs broken	0.007	0.019	0.032	0.378	0.706	−0.030	0.044

CI, confidence interval; GCS, Glasgow coma scale; ISS, injury severity score; SSRF, surgical stabilization of rib fracture. All variables with *p* < 0.05 in univariate analysis are included in a multivariate regression model of natural log-transformed length of hospital stay.

**Table 8 jpm-12-01844-t008:** Multivariate logistic regression analysis for nonprocedure-related pulmonary complications of SSRF (+) group.

Variables	Univariate Model Results			Multivariate Model Results		
	OR	95% CI	*p*	OR	95% CI	*p*
Age, y	1.050	1.006–1.095	0.025	1.097	1.025–1.173	0.007
GCS	0.662	0.541–0.810	<0.001	0.648	0.499–0.843	0.001
Flail chest requiring mechanical ventilation	8.904	2.707–29.286	<0.001	4.643	0.748–28.827	0.135
Number of surgically fixated rib fractures	1.416	1.012–1.982	0.042	1.514	0.667–3.438	0.322
Operation time	1.009	1.001–1.017	0.023	1.004	0.990–1.018	0.600
Blood loss during SSRF	1.002	1.000–1.005	0.018	1.002	0.999–1.004	0.176
Number of ribs broken	1.099	0.903–1.337	0.347	0.734	0.444–1.211	0.226
Video-assisted vs. Thoracotomy	0.569	0.176–1.833	0.344	0.232	0.038–1.417	0.113

CI, confidence interval; GCS, Glasgow coma scale; ISS, injury severity score; SSRF, surgical stabilization of rib fracture. All variables with *p* < 0.05 in univariate analysis are included in a multivariate regression model of natural log-transformed nonprocedure-related pulmonary complications.

**Table 9 jpm-12-01844-t009:** Postpropensity scores matching demographic and perioperative variables of the patients.

	SSRF (−) N = 106	SSRF (+) N = 106	*p*
Age, y	58.5 (43.5–70.0)	56.5 (46.8–64.3)	0.37
Male/Female	69/37	64/42	0.48
Transferal	32 (30.2)	35 (33.0)	0.66
Charlson Comorbidity Index	2.0 (0.0–4.0)	2.0 (0.0–3.0)	0.50
Trauma mechanism			0.50
Vehicle-to-vehicle collision	59 (55.7)	68 (64.2)	
Vehicle-to-pedestrian collision	4 (3.8)	5 (4.7)	
Single-vehicle collision	25 (23.6)	17 (16.0)	
Fall	18 (17.0)	16 (15.1)	
Crush	0 (0.0)	0 (0.0)	
Assault	0 (0.0)	0 (0.0)	
Glasgow Coma Scale	15.0 (15.0–15.0)	15.0 (15.0–15.0)	0.95
Injury Severity Score	17.0 (13.0–22.5)	17.0 (13.0–22.0)	0.30
Head/Neck			
Median (IQR)	0.0 (0.0–2.0)	0.0 (0.0–2.0)	0.51
AIS ≥ 3	23 (21.7)	24 (22.6)	0.87
Face			
Median (IQR)	0.0 (0.0–0.0)	0.0 (0.0–0.0)	0.34
AIS ≥ 3	1 (0.9)	0 (0.0)	>0.99
Thorax			
Median (IQR)	3.0 (3.0–3.0)	3.0 (3.0–3.0)	0.29
AIS ≥ 3	99 (93.4)	106 (100.0)	0.02
Abdomen			
Median (IQR)	0.0 (0.0–2.0)	0.0 (0.0–2.0)	0.49
AIS ≥ 3	15 (14.2)	12 (11.3)	0.54
Extremity			
Median (IQR)	2.0 (0.0–2.0)	2.0 (0.0–2.0)	0.82
AIS ≥ 3	20 (18.9)	11 (10.4)	0.12
External			
Median (IQR)	0.0 (0.0–1.0)	0.0 (0.0–0.0)	0.60
AIS ≥ 3	0 (0.0)	0 (0.0)	NA
Number of ribs broken	5.0 (3.0–8.0)	6.0 (5.0–8.0)	0.02
Fractured side			0.24
Unilateral	88 (83.0)	94 (88.7)	
Bilateral	18 (17.0)	12 (11,3)	
Presence of a flail segment radiologically	35 (33.0)	35 (33.0)	NA
Requiring mechanical ventilation	10 (9.4)	6 (5.7)	0.30
Associated intrathoracic injury			
Lung contusion/laceration	50 (47.2)	39 (36.8)	0.13
Pneumothorax	44 (41.5)	57 (53.8)	0.07
Hemothorax	72 (67.9)	83 (78.3)	0.09
Cardiac injury	3 (2.8)	1 (0.9)	0.62
Great vessels	2 (1.9)	1 (0.9)	>0.99
Soft tissue	13 (12.3)	18 (17.0)	0.33
Concurrent sternal fracture	2 (1.9)	4 (3.8)	0.69
Concurrent ipsilateral clavicular fracture	38 (35.8)	45 (42.5)	0.33
Concurrent ipsilateral scapular fracture	17 (16.0)	20 (18.9)	0.59

Flail chest, the presence of three or more contiguous ribs fractured in two or more places; fracture fixation ratio, total fixated ribs divided by total fractured ribs; SSRF, surgical stabilization of rib fractures.

**Table 10 jpm-12-01844-t010:** Postpropensity scores matching outcome variables of the patients.

	SSRF (−) N = 106	SSRF (+) N = 106	*p*
Ventilator days, d	0.0 (0.0–9.8)	2.0 (0.0–6.0)	0.877
	N = 52	N = 52	
Length of ICU stay, d	3.0 (0.0–7.5)	3.0 (0.0–8.0)	0.914
	N = 69	N = 69	
Length of hospital stay, d	8.0 (5.0–20.0)	11.5 (8.0–18.0)	0.195
Time from trauma to ambulation, d	4.0 (3.0–10.0)	3.0 (1.0–6.0)	0.002
Nonprocedure-related pulmonary complication	18 (17.0)	7 (6.6)	0.019
Tracheostomy	4 (3.8)	0 (0.0)	0.121
Mortality	1 (0.9)	0 (0.0)	>0.999

SSRF, surgical stabilization of rib fractures.

## Data Availability

The data that support the findings of this study are available from National Cheng Kung University Hospital, but restrictions apply to the availability of these data, which were used under license for the current study, and so are not publicly available. Data are, however, available from the authors upon reasonable request and with permission of correspondence.

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
