# Peer review of "Tailored Surgical Stabilization of Rib Fractures Matters More Than the Number of Fractured Ribs"

_jpm, 2022, doi:10.3390/jpm12111844_

Round 1
Reviewer 1 Report
This is a well designed study on the use of surgical stabilization of rib fractures. The authors have retrospectively included 174 consecutive patients to assess predictors of poor outcome.
1. Please explain better your exclusion criteria.
2. I would like to see the rib number (1st, second etc) to be included in the linear regression analyses
3. Why did you do regression only for a subset of your outcome variables?
4. What does the P-value on Table 4 represent?
5. Please use all other variables for your analysis or justify why you didnt use them.
Author Response
Thanks for rigorous advice.
Point 1. Please explain better your exclusion criteria.
Response 1: This is an important question. A number of exclusion criteria included trauma victims without fractured ribs, rib fracture patients not receiving SSRF, rib fractures not resulting from trauma, and patients without SSRF indications.
Point 2. I would like to see the rib number (1st, second etc) to be included in the linear regression analyses I would like to see the rib number (1st, second etc) to be included in the linear regression analyses
Response 2. The rib number has been included as one variable in multiple linear regression analyses in Table 5, and Table 6.
Point 3. Why did you do regression only for a subset of your outcome variables?
Response 3. In this study, outcomes included pulmonary complications, ventilator days, length of hospital and ICU stays, and time from trauma to ambulation. Our patients, however, suffered from multi-trauma; therefore, length of hospital stay and time from trauma to ambulation are possibly influenced by several factors, such as concomitant long bone fractures, and poor consciousness resulting from head injury, which does not truly reflect the benefit of SSRF. This is why we presented these outcomes without analyzing them multivariably.
Point 4. What does the P-value on Table 4 represent?
Response 4. Mortality rate and incidence of pulmonary complications increased significantly when six or more ribs were fractured. Therefore, in Table 4, we compared patients with less than six fractured ribs with those with more than six fractured ribs without adjustment. The P-value indicates that patients with more than six fractured ribs had worse outcomes in both pulmonary complications and time-related outcomes. Our analysis of multiple linear regression models indicated, however, that fractured rib number was not associated with adverse outcomes after stabilization.
Point 5. Please use all other variables for your analysis or justify why you didnt use them.
Response 5. As reviewer’s suggestion, we entered into multiple linear regression those factors with P-value <0.2. Based on established factors associated with outcomes in the literature and key factors addressed in this study, fracture numbers and surgical approaches were also included in the analyses. However, because of limited pulmonary outcomes, we applied backward stepwise analysis on multivariate logistic regression. In general, the results remained the same.

Reviewer 2 Report
Authors studied the outcomes in patients with rib fractures, they retrospectively reviewed outcomes of patients receiving tailored SSRF and tried to identify factors associated with poor surgical outcomes.
While the study is well conducted, the number of patients is more than sufficient and statistical analysis is pertinent, little is known about the average patients follow up and if they underwent physical therapy after stabilization or not.
Author should talk more about what I have stated above.
Authors should cite this article:
La Padula S, Hersant B, Meningaud JP. Intraoperative use of indocyanine green
angiography for selecting the more reliable perforator of the anterolateral thigh flap: A
comparison study. Microsurgery. 2018 Oct;38(7):738-744.
in which the importance of physical therapy after trauma is well illustrated.
Author Response
Point 1. While the study is well conducted, the number of patients is more than sufficient and statistical analysis is pertinent, little is known about the average patients follow up and if they underwent physical therapy after stabilization or not.
Response 1. Your comment is greatly appreciated. Since there was no mortality, all patients were followed up for at least 1 month postoperatively. Median (interquartile range) follow-up time was 22.0 (8.4-36.0) months.
We routinely consult physiotherapists for all trauma patients at our Trauma team. However, it was up to the specialty of the physiotherapists to decide what kind of rehabilitation they needed.
Point 2. Authors should cite this article:
La Padula S, Hersant B, Meningaud JP. Intraoperative use of indocyanine green angiography for selecting the more reliable perforator of the anterolateral thigh flap: A comparison study. Microsurgery. 2018 Oct;38(7):738-744.
in which the importance of physical therapy after trauma is well illustrated.
Response 2. I appreciate the reference you provided.

Reviewer 3 Report
I would like to thank the authors for submitting their research on this interesting topic. Surgical stabilization of rib fractures has been intensively discussed in the literature. It remains unclear which new aspect is added to this topic by this paper. The method has some serious flaws (inclusion seems to depend on insurance status, GCS is stated as 15 for all patients in table 2, however later on the author refer to patients with lower GCS), especially the authors failed (which is also mentioned in the limitations section) to establish a control group. I would advise the authors to continue their work and compare their results to a suitable control group and then submit a new paper.
Author Response
Point 1. It remains unclear which new aspect is added to this topic by this paper.
Response 1. Thanks for serious comments. We all know that the number of fractured ribs correlates with the severity of thoracic trauma, and that surgical stabilization of rib fractures (SSRF) can reduce ventilator days and pain scores. But little is said about “Is the number of fractured ribs still important after SSRF?” This article suggests that there is no effect of the number of fractured ribs post-SSRF.
Point 2. The method has some serious flaws (inclusion seems to depend on insurance status, GCS is stated as 15 for all patients in table 2, however later on the author refer to patients with lower GCS), especially the authors failed (which is also mentioned in the limitations section) to establish a control group.
Response 2. The GCS stated in table 2 may be misleading. A left-skewed distribution results in 15 (15-15) as the median and interquartile range. 144 patients (82.8%) had GCS scores of 15. There was a mean (standard deviation) GCS score of 14.3 (2.0), ranging from 3 to 15. Because SSRF is not covered by national health insurance in our country, each rib cost around 2000 USD, resulting in selection bias. Consequently, many patients might not receive SSRF due to price rather than severity, which also impacts the choice of appropriate control group. In the future, we will attempt to manage this economic issue by designing an appropriate matched study.

Reviewer 4 Report
Dear Authors
Congratulations on a wonderful study that is appropriately planned and executed. I have some concerns with regard to the presentation of the materials such as tables.
Introduction can be improved to comment on the need and importance of the subject being analyzed.
If the authors could have compared this data with the data from the same hospital without the algorithmic approach before its implementation then it would serve as a benchmark to validate its importance and need and effectiveness.
The contents of the demographics and preop variables table can be appropriately grouped to make it more presentable to the readers.
Author Response
Point 1. Introduction can be improved to comment on the need and importance of the subject being analyzed.
Response 1. We all know that the number of fractured ribs correlates with the severity of thoracic trauma, and that surgical stabilization of rib fractures (SSRF) can reduce ventilator days and pain scores. But little is said about “Is the number of fractured ribs still important after SSRF?” This article suggests that there is no effect of the number of fractured ribs post-SSRF.
Point 2. If the authors could have compared this data with the data from the same hospital without the algorithmic approach before its implementation then it would serve as a benchmark to validate its importance and need and effectiveness.
Response 2. Comparing current data with data without algorithmic approach is a great idea, but we had to be honest about the fact that electronic medical records started at our institute around 2009, so information before 2010 might not be complete. For comparison, perhaps we could seek another medical center with the same level but without the algorithmic approach.
Point 3. The contents of the demographics and preop variables table can be appropriately grouped to make it more presentable to the readers.
Response 3. The demographics and preoperative variables tables have been reorganized to make them easier to read.

Round 2
Reviewer 1 Report
I am still not satisfied by the way variables were included in the regression analysis. The authors have to check with univariate regression all variables. If P<0.05 they have to be included in the multivariate model no matter what. All results of univariate and multivariate logistic regression should be reported in a table. Statements like "in general the results remained the same...." are not appropriate for a research manuscript.
Author Response
Author response #1: Your suggestions were greatly appreciated. Based on the reviewer's comments, we entered all variables with P <0.05 in univariate analysis to multivariate analysis. Additionally, detailed univariate results are provided below, as well as in Table S4.
P value of univariate analyses of SSRF (+) group |
|
||||
Logistic regression for NPRCs |
Linear regression for log-ventilator days |
Linear regression for log-ICU stay |
Linear regression for log-hospital stay |
|
|
Age at time of surgery, y |
0.03 |
0.65 |
0.87 |
0.60 |
|
Sex (female vs. male) |
0.20 |
0.54 |
0.35 |
0.65 |
|
Transferal |
0.34 |
0.79 |
0.62 |
0.44 |
|
Charlson comorbidity index |
0.14 |
0.33 |
0.80 |
0.98 |
|
Non-car accidents vs. Car accidents |
0.42 |
0.15 |
0.87 |
0.45 |
|
Glasgow coma scale |
<0.01 |
<0.01 |
<0.01 |
<0.01 |
|
Injury severity score |
0.08 |
0.01 |
<0.01 |
<0.01 |
|
Number of ribs broken |
0.35 |
0.20 |
0.04 |
<0.01 |
|
Fractured side (bilateral vs. unilateral) |
>0.99 |
0.49 |
0.04 |
0.04 |
|
Presence of a flail segment radiologically |
0.12 |
0.29 |
0.33 |
<0.01 |
|
Requiring mechanical ventilation |
<0.01 |
<0.01 |
<0.01 |
<0.01 |
|
Associated intrathoracic injury |
|
|
|
|
|
Lung contusion/laceration |
>0.99 |
0.45 |
<0.01 |
<0.01 |
|
Pneumothorax |
0.28 |
0.37 |
0.14 |
0.22 |
|
Hemothorax |
0.47 |
0.86 |
0.91 |
0.75 |
|
Cardiac injury |
>0.99 |
0.62 |
0.87 |
0.96 |
|
Great vessels injury |
>0.99 |
0.53 |
0.39 |
0.14 |
|
Soft tissue injury |
0.57 |
0.62 |
0.56 |
0.05 |
|
Concurrent sternal fracture |
>0.99 |
0.75 |
0.86 |
0.46 |
|
Concurrent ipsilateral clavicular fracture |
0.24 |
0.28 |
0.72 |
0.74 |
|
Concurrent ipsilateral scapular fracture |
0.61 |
0.27 |
0.29 |
0.42 |
|
Time from trauma to rib fixation, d |
0.34 |
<0.01 |
<0.01 |
<0.01 |
|
Number of surgically fixated rib fractures |
0.04 |
0.09 |
0.17 |
<0.01 |
|
Fracture fixation ratio |
0.27 |
0.89 |
0.47 |
0.27 |
|
Double lumen vs. Single lumen intubation |
0.17 |
0.13 |
0.06 |
0.02 |
|
Approach side (bilateral vs. unilateral) |
>0.99 |
0.45 |
0.29 |
0.12 |
|
Video-assisted mini-thoracotomy vs. Thoracotomy |
0.34 |
0.04 |
0.04 |
0.13 |
|
Concomitant surgery for other injuries |
0.36 |
0.20 |
0.75 |
0.55 |
|
Operation time, min |
0.02 |
0.05 |
0.20 |
<0.01 |
|
Blood loss during SSRF |
0.02 |
0.01 |
0.01 |
<0.01 |
|
NPRCs, Non-Procedural Postoperative Pulmonary Complications; SSRF, surgical stabilization of rib fractures |
Reviewer 3 Report
I would like the authors for submitting a revised version of their paper. However, the study design (no control group) does in my opinion still not fit the high quality standards of this journal. I thus recommend the authors to include a control group (matched pair design) and compare the outcomes of this control group with their interventional group.
Author Response
Author response #3: Thank you for your insightful comment. Our team reviewed rib fracture patients without SSRF between 2019 and 2020. Propensity score matching was used to compare their outcomes. Tables of post-propensity score matching perioperative variables and outcomes of the patients are provided below and are included in the manuscript.
Post-propensity score matching demographic and perioperative variables of the patients |
|||
SSRF (-) N = 106 |
SSRF (+) N = 106 |
P |
|
Age, y |
58.5 (43.5-70.0) |
56.5 (46.8-64.3) |
0.37 |
Male/Female |
69/37 |
64/42 |
0.48 |
Transferal |
32 (30.2) |
35 (33.0) |
0.66 |
Charlson Comorbidity Index |
2.0 (0.0-4.0) |
2.0 (0.0-3.0) |
0.50 |
Trauma mechanism |
|
|
0.50 |
Vehicle-to-vehicle collision |
59 (55.7) |
68 (64.2) |
|
Vehicle-to-pedestrian collision |
4 (3.8) |
5 (4.7) |
|
Single-vehicle collision |
25 (23.6) |
17 (16.0) |
|
Fall |
18 (17.0) |
16 (15.1) |
|
Crush |
0 (0.0) |
0 (0.0) |
|
Assault |
0 (0.0) |
0 (0.0) |
|
Glasgow Coma Scale |
15.0 (15.0-15.0) |
15.0 (15.0-15.0) |
0.95 |
Injury Severity Score |
17.0 (13.0-22.5) |
17.0 (13.0-22.0) |
0.30 |
Head/Neck |
|
|
|
Median (IQR) |
0.0 (0.0-2.0) |
0.0 (0.0-2.0) |
0.51 |
AIS ≥3 |
23 (21.7) |
24 (22.6) |
0.87 |
Face |
|
|
|
Median (IQR) |
0.0 (0.0-0.0) |
0.0 (0.0-0.0) |
0.34 |
AIS ≥3 |
1 (0.9) |
0 (0.0) |
>0.99 |
Thorax |
|
|
|
Median (IQR) |
3.0 (3.0-3.0) |
3.0 (3.0-3.0) |
0.29 |
AIS ≥3 |
99 (93.4) |
106 (100.0) |
0.02 |
Abdomen |
|
|
|
Median (IQR) |
0.0 (0.0-2.0) |
0.0 (0.0-2.0) |
0.49 |
AIS ≥3 |
15 (14.2) |
12 (11.3) |
0.54 |
Extremity |
|
|
|
Median (IQR) |
2.0 (0.0-2.0) |
2.0 (0.0-2.0) |
0.82 |
AIS ≥3 |
20 (18.9) |
11 (10.4) |
0.12 |
External |
|
|
|
Median (IQR) |
0.0 (0.0-1.0) |
0.0 (0.0-0.0) |
0.60 |
AIS ≥3 |
0 (0.0) |
0 (0.0) |
NA |
Number of ribs broken |
5.0 (3.0-8.0) |
6.0 (5.0-8.0) |
0.02 |
Fractured side |
|
|
0.24 |
Unilateral |
88 (83.0) |
94 (88.7) |
|
Bilateral |
18 (17.0) |
12 (11,3) |
|
Presence of a flail segment radiologically |
35 (33.0) |
35 (33.0) |
NA |
Requiring mechanical ventilation |
10 (9.4) |
6 (5.7) |
0.30 |
Associated intrathoracic injury |
|
|
|
Lung contusion/laceration |
50 (47.2) |
39 (36.8) |
0.13 |
Pneumothorax |
44 (41.5) |
57 (53.8) |
0.07 |
Hemothorax |
72 (67.9) |
83 (78.3) |
0.09 |
Cardiac injury |
3 (2.8) |
1 (0.9) |
0.62 |
Great vessels |
2 (1.9) |
1 (0.9) |
>0.99 |
Soft tissue |
13 (12.3) |
18 (17.0) |
0.33 |
Concurrent sternal fracture |
2 (1.9) |
4 (3.8) |
0.69 |
Concurrent ipsilateral clavicular fracture |
38 (35.8) |
45 (42.5) |
0.33 |
Concurrent ipsilateral scapular fracture |
17 (16.0) |
20 (18.9) |
0.59 |
Flail chest, the presence of three or more contiguous ribs fractured in two or more places; fracture fixation ratio, total fixated ribs divided by total fractured ribs; SSRF, surgical stabilization of rib fractures |
Post-propensity score matching outcome variables of the patients |
|||
SSRF (-) N = 106 |
SSRF (+) N = 106 |
P |
|
Ventilator days, d |
0.0 (0.0-9.8) N = 52 |
2.0 (0.0-6.0) N = 52 |
0.877 |
Length of ICU stay, d |
3.0 (0.0-7.5) N = 69 |
3.0 (0.0-8.0) N = 69 |
0.914 |
Length of hospital stay, d |
8.0 (5.0-20.0) |
11.5 (8.0-18.0) |
0.195 |
Time from trauma to ambulation, d |
4.0 (3.0-10.0) |
3.0 (1.0-6.0) |
0.002 |
Nonprocedure-related pulmonary complication |
18 (17.0) |
7 (6.6) |
0.019 |
Tracheostomy |
4 (3.8) |
0 (0.0) |
0.121 |
Mortality |
1 (0.9) |
0 (0.0) |
>0.999 |
SSRF, surgical stabilization of rib fractures |